# Identification of Aichivirus in a Pet Rat (*Rattus norvegicus*) in Italy

**DOI:** 10.3390/ani14121765

**Published:** 2024-06-11

**Authors:** Flora Alfano, Maria Gabriella Lucibelli, Francesco Serra, Martina Levante, Simona Rea, Amalia Gallo, Federica Petrucci, Alessia Pucciarelli, Gerardo Picazio, Marina Monini, Ilaria Di Bartolo, Dario d’Ovidio, Mario Santoro, Esterina De Carlo, Giovanna Fusco, Maria Grazia Amoroso

**Affiliations:** 1Istituto Zooprofilattico Sperimentale del Mezzogiorno, 80055 Portici, Italy; mariagabriella.lucibelli@izsmportici.it (M.G.L.); francesco.serra@izsmportici.it (F.S.); martina.levante@izsmportici.it (M.L.); simona.rea@izsmportici.it (S.R.); amalia.gallo@izsmportici.it (A.G.); alessia.pucciarelli@izsmportici.it (A.P.); gerardo.picazio@izsmportici.it (G.P.); direzionesanitaria@izsmportici.it (E.D.C.); giovanna.fusco@izsmportici.it (G.F.); mariagrazia.amoroso@izsmportici.it (M.G.A.); 2Istituto Superiore di Sanità, 00161 Rome, Italy; marina.monini@iss.it (M.M.); ilaria.dibartolo@iss.it (I.D.B.); 3Istituto di Gestione della Fauna, 80126 Napoli, Italy; dariodovidio@yahoo.it; 4Stazione Zoologica Anton Dohrn, 80122 Napoli, Italy; mario.santoro@szn.it

**Keywords:** aichivirus, *Rattus norvegicus*, exotic pets, kobuvirus

## Abstract

**Simple Summary:**

Dozens of species of small exotic mammals, such as chinchillas, golden hamsters, Java squirrels, Mongolian gerbils, mice, rats, African hedgehogs, and sugar gliders, are kept as pets globally. These exotic companion mammals are potential reservoirs for maintaining and transmitting zoonotic pathogens. Using molecular assays, our study aimed to investigate the occurrence of some potential zoonotic viruses in various small exotic mammals kept as pets in Italy. Our findings show that aichivirus (AiV) was further characterized as murine kobuvirus-1 in a rat (*Rattus norvegicus*). To the best of our knowledge, this is the first study reporting the detection of AiV in rodents in Italy. Our results also revealed the absence of other viruses investigated while highlighting the importance of ongoing monitoring of infectious agents in these increasingly common pets to prevent the spread of new potential zoonotic pathogens.

**Abstract:**

We investigated the occurrence of eight potential zoonotic viruses in 91 exotic companion mammals from pet shops in southern Italy via real-time PCR and end-point PCR. The animals were screened for aichivirus, sapovirus, astrovirus, hepatitis A, noroviruses (GI and GII), rotavirus, circovirus, and SARS-CoV-2. Among the nine species of exotic pets studied, only one rat tested positive for aichivirus. The high sequence similarity to a murine kobuvirus-1 strain previously identified in China suggests that the virus may have been introduced into Italy through the importation of animals from Asia. Since exotic companion mammals live in close contact with humans, continuous sanitary monitoring is crucial to prevent the spread of new pathogens among domestic animals and humans. Further investigations on detecting and typing zoonotic viruses are needed to identify emerging and re-emerging viruses to safeguard public health.

## 1. Introduction

The global demand for non-conventional pets has greatly increased in recent decades. It has been estimated that in Italy alone, there are about 30 million ornamental fishes, 1.4 million reptiles, 12.9 million ornamental birds, and almost 1.8 million small mammals, such as chinchillas, guinea pigs, ferrets, prairie dogs, meerkats, golden hamsters, Java squirrels, Mongolian gerbils, mice, rats, rabbits, African hedgehogs, sugar gliders, and others [1,2,3,4,5,6]. The close social interaction between animals and humans provides a strong motivation to investigate the virome composition of non-conventional pets, as these animals could represent a significant source of infection due to little-known or even unknown virus agents [2,7,8]. Furthermore, the illegal trafficking of these pets contributes to increasing the risk of introducing new pathogens into free areas that could be transmitted to other animals and/or humans. Hundreds of non-conventional pets are destined for the black market, whose turnover is second only to drug trafficking (Zoomafia Observatory Anti-Vivisection League). Animals purchased on the black market are not subject to any health checks and may originate from areas where viral pathogens are endemic [4,9,10]. Open borders and a constant increase in human and animal movement could trigger the rapid emergence and spread of new pathogens globally. Hence, identifying new viral agents, especially in animal species in close contact with humans, is of fundamental importance for assessing the zoonotic risk.

The intimate connections between the health of humans, their pets, and the environment are recognized by the “One Health” approach. The importance of identifying and understanding these connections to control and prevent disease is reminded by the emergence of infectious diseases such as COVID-19. Consequently, there is an increasing research focus on viruses that cause disease in companion animals, which consequently affect human health [11,12].

Zoonotic pathogens may infect exotic pets, particularly rodents, and they may harbor a disproportionate number of them [13]. The virology of exotic companion mammals used as pets is unknown because of their recent introduction into households. Several species may pose a potential zoonotic hazard to humans.

Referring to the literature, we investigated the following viruses: rotavirus, hepatitis, sapoviruses, kobuviruses, and circovirus. Rotavirus and hepatitis have been identified in animals such as rats, mice, and rabbits [12,14,15,16]. Sapoviruses, kobuviruses [4], and circovirus [17] have also been identified in rats. A novel picornavirus has been detected in fecal samples of rabbits, demonstrating that they may act as hosts of kobuviruses [18]. Astrovirus has been identified in mice [19].

Further studies are necessary to better understand pet viral infections and recognize their zoonotic potential.

Our study aimed to assess the circulation of some potential zoonotic viruses, including aichivirus (A and B AiV), human sapovirus (SaV), human astrovirus (AstV), hepatitis A virus (HAV), human noroviruses (NoVGI and NoVGII), group A rotavirus (RVA), porcine circovirus (PCV), and SARS-CoV-2, in exotic companion mammals imported into Italy from various EU and non-EU countries. The data obtained will expand the current limited knowledge on this topic.

## 2. Materials and Methods

### 2.1. Animals Investigated

Between 2021 and 2022, organs and tissues obtained at necropsy from 91 exotic companion mammals kept as pets in Italy, most of which died from unknown causes, were collected and successively analyzed for selected zoonotic viruses. A total of 637 samples from the brain, lungs, heart, liver, spleen, kidney, and intestine were sterilely collected and sent to the laboratory of the Istituto Zooprofilattico Sperimentale del Mezzogiorno for virological investigations, where they were immediately analyzed or stored at −80 °C before DNA/RNA extraction.

The animals investigated included 16 golden hamsters (*Mesocricetus auratus*), 6 Java squirrels (*Callosciurus notatus*), 2 Mongolian gerbils (*Meriones unguiculatus*), 11 guinea pigs (*Cavia porcellus*), 20 mice (*Mus musculus*), 21 rats (*Rattus norvegicus*), 12 rabbits (*Oryctolagus cuniculus*), 2 African hedgehogs (*Atelerix albiventris*), and 1 sugar glider (*Petaurus breviceps*). Most animals came from veterinary clinics, pet shops, and private breeders and were bred and marketed in Italy. The African hedgehogs first arrived in central Europe (the Czech Republic) and then were brought to Italy by European importers. The squirrels were directly imported from Malaysia and Thailand. Most animals arrived in large loads and died from stressful conditions during transportation.

### 2.2. Nucleic Acids Extraction

For nucleic acid extraction, 25 mg of each organ was homogenized by a Tissue Lyser (Qiagen GmbH, Hilden, Germany) in 2 mL Eppendorf safe-lock tubes containing 1 mL of phosphate-buffered saline solution (PBS) as described previously [20]. Extraction was carried out on 200 µL of organ homogenate by using a QIAsymphony automated extraction system (Qiagen GmbH, Hilden, Germany) with a DSP Virus/Pathogen Mini kit (Qiagen GmbH, Hilden, Germany) according to the manufacturer’s instructions. A sample with 200 µL of PBS instead of homogenate was used as a negative process control (NPC).

### 2.3. Viral Recovery with Murine Norovirus

Before sample extraction, murine norovirus (MNV-1) [21] was added to each sample to calculate viral recovery and the presence of PCR inhibitors (external process control, EPC). In particular, murine macrophage-like RAW264.7 cells grown in Dulbecco’s Modified Eagle’s Medium (D-MEM) were inoculated with MNV-1 (MNV-IT1 Acc. no. KR349276). After three days, the medium solution was removed. Every 3 days, the medium was replaced and used to infect the new RAW264.7 cell monolayer. After six repetitions of these steps, the viral titer was calculated by end-point dilution, and a final stock of 10^7^ PFU mL^−1^ was obtained. In total, 5 µL of viral stock (10^7^ PFU mL^−1^) was added to each sample before extraction. Detection of EPC was performed by real-time PCR, as indicated in the literature [22]. The sample extraction efficiency (R) was calculated by comparing the Ct value of MNV-1 in the sample with the Ct value of MNV-1 in the viral stock added to the samples [23]. When an acceptable recovery rate was obtained (R > 1), the results were analyzed as follows: if the threshold cycle (Ct) of the EPC in the eluted sample was comparable to that of the EPC in the NPC, the sample was analyzed as undiluted. If the difference between the two Cts was at least 3 or a multiple of 3, all the analyses were carried out on the sample diluted at a ratio of 1:10 or more [20].

### 2.4. Detection of Viral Nucleic Acid by Biomolecular Analysis

Extracted nucleic acid was analyzed by real-time polymerase chain reaction (PCR) and real-time reverse transcription PCR (RT-PCR) for the presence of the following eight viruses: AiV-A, AiV-B, human SaV, human AstV, HAV, human NoVGI and GII, RVA, PCV-2, PCV-1, and SARS-CoV-2, with primers and probes indicated in Appendix A.

Real-time virus detection was performed in a final reaction volume of 25 μL with 5 μL of nucleic acid extract. The circovirus viral genome was investigated by real-time PCR using a Quantitect Real-Time PCR detection kit (Qiagen, Hilden, Germany). The presence of AiV, SaV, AstV, HAV, NoVGI, NoVGII, and RVA genomic sequences was examined by real-time RT-PCR using an AgPath-ID™ one-step RT-PCR kit (Thermo Fisher Scientific, Waltham, MA, USA). All reactions were carried out in single runs on a Quant Studio 5 system (Thermo Fisher Scientific, Waltham, MA, USA) using primers (Tema Ricerca-Castenaso, Bologna, Italy) and probes (Thermo Fisher Scientific, Waltham, MA, USA) specific for the viruses tested (see Appendix A) and following the thermal profiles indicated in the literature (Appendix A). The positive controls employed are described herein. The Istituto Superiore di Sanità, Rome, Italy, provided HAV-, NoVGI-, NoVGII-, and RVA-positive plasmid controls. AstV and SaV Q standards were purchased from ceeramTOOLS (bioMérieux). An AiV-positive control (a human-stool-positive sample) was provided by the Centre National de Référence des Virus Entériques (Dijon Cedex, France). PCV was obtained from a commercial kit (VetMAX Porcine PCV2 Quant kit).

The analysis for SARS-COV-2 detection was carried out using the following different diagnostic kits to evaluate their efficiency in the case of positive samples: (1) RADI COVID-19 Detection Kit—Menarini Diagnostics; (2) ARGENE^®^ SARS-COV-2 R-GENE^®^ BIOMÉRIEUX; (3) TaqMan™ 2019nCoV Assay Kit v1—Thermo Fisher Scientific; (4) Allplex™ SARS-CoV-2 Assay—Seegene; and (5) TaqPath COVID-19 CE-IVD RT-PCR Kit (Thermo Fisher Scientific) and SARS-CoV-2 Droplet digital polymerase chain reaction (dd-PCR).

### 2.5. Aichivirus Characterization

To further characterize the aichivirus-positive amplicon identified, end-point RT-PCR was performed using a one-step RT-PCR kit (Qiagen) followed by a nested PCR using the VP1-F/VP1-R primers pairs, which amplifies a fragment of approximately 296 bp within the VP1 (capsid protein) [24].

The PCR product with the expected size was then purified through an enzymatic reaction (USB^®^ ExoSAP-IT, Affymetrix UK Ltd., Altrincham, UK) and subjected to Sanger sequencing (Eurofins Genomic, Milan, Italy).

The sequence named MuKb_PO207_ITA23 was submitted to the NCBI database under the accession number OR828459.

The 200 bp VP1 fragment of the strain MuKb_PO207_ITA23 was aligned with sequences retrieved from the NCBI database using MEGA X software version 10 [25]. The maximum likelihood phylogenetic tree was built using model TN93+G+I, as suggested by the model test method, with 1000× bootstraps (MEGA X).

## 3. Results

The age categories (newborn, young, and adult) and sex of the exotic companion mammals investigated are listed in Appendix A.

Each animal’s organs (brain, lungs, heart, liver, spleen, kidneys, and intestine) underwent single nucleic acid extraction and subsequent investigation (by single reactions) of the eight viruses under study. AstV, HAV, NoVGI, NoVGII, PCV, RVA, SaV, and SARS-CoV-2 were not detected in any of the organs of the 91 exotic companion mammals investigated.

Among the organs and tissues collected from 91 exotic companion mammals, only the intestine of an adult male rat tested positive for aichivirus, showing a threshold cycle of 30. The positive sample underwent end-point PCR for the partial amplification of the VP1 gene, followed by sequencing and phylogenetic analysis. The obtained partial VP1 sequence was compared with the public sequence database using BlastN, and the results confirmed that the AiV belonged to the species Aichivirus A, specifically the genus murine kobuvirus (MuKV). The characteristics of the selected sequences used for constructing the phylogenetic tree are reported in Appendix A. Our sequence exhibited the highest level of nucleotide identity, reaching 96% (100% aa. Id.) with a strain of murine kobuvirus 1 called MuKV/YN27/CHN (MW292480) that was identified in China (see Figure 1).

The detected strain belonged to a group of strains exclusively found in China, while strains identified in the USA and Hungary were not as closely related (<90%nd. id). The strain displayed a limited identity with human strains (85% aa. id).

## 4. Discussion

Among the nine species of exotic companion mammals we studied to investigate the circulation of potentially zoonotic viruses, only one rat tested positive for AiV. The partial sequence of the aichivirus found in the intestine of the rat showed the closest identity (96%) with a murine kobuvirus 1 strain called MuKV/YN27/CHN (MW292480), identified in a rat in China [5,26].

According to the International Committee for Virus Taxonomy, AiV is a single-stranded RNA virus belonging to the order Picornavirales, family Picornaviridae, and genus Kobuvirus [27] isolated for the first in a human with oyster-related non-bacterial diarrhea in Japan [27].

There are six recognized species of aichivirus (named from A to F) and three tentative species in the genus Kobuvirus, namely caprine kobuvirus, Norway rat kobuvirus, and bat kobuvirus; a viral genome related to aichivirus was retrieved using a metagenomic approach from feces of a bat [28,29].

The species AiV A contains six types: aichivirus, murine kobuvirus, feline kobuvirus (FeKoV), roller kobuvirus, canine kobuvirus, and Kathmandu sewage kobuvirus [30]. The other species also exhibit a considerable degree of variability, three of which have been identified in aichivirus B and two of which have been identified in aichivirus C, aichivirus D, and aichivirus F. So far, three genotypes of AiV 1, genotypes A, B, and C, have been discovered [31]. Genotype A is mostly found in Japan and Europe, while genotype B is commonly found in Brazil and several countries of Asia, except Japan [32,33,34]. Genotype C has only been detected in Africa [32].

Regarding the strain found here, MuKV is distinct from human strains despite all belonging to aichivirus A [35]. However, the obtained fragment within VP1 does not allow for an in-depth analysis of genomic variability. Notably, the available sequences are predominantly from China, influencing the comparison results.

All the rats analyzed in this study were born in Italy and bred to be sold as pets. While in the pet shops, some came into contact with other rodents from non-EU countries like Malaysia and Thailand. Since, according to the literature, no other AiV has been described in pet rats in Italy, other rodents, or other exotic companion mammals, we hypothesized that the virus was introduced to the country by importing infected rodents/mammals from Asia. As suggested for other viral agents [9,28], the AiV-positive rat detected in this study could have come into close contact (maybe in pet shops) with our positive rat.

Previous detection of AiV in rats worldwide includes reports from China [26,36,37] and some records from Asia and America from the feces of rodents [4,28,36], while in Europe, MuKVs were isolated in Hungary [1].

Members of the genus *Kobuvirus*, genetically closely related to human AiVs, have been identified in Italy from a wide range of domestic and wild animals, suggesting a mutual exchange of viruses [38,39,40,41,42,43,44,45,46].

None of the other viruses analyzed (AstV, HAV, SaV, NoVGI, NoVGII, PCV, RVA, and SARS-CoV-2) tested positive in our analysis. These data reassure us about the absence of almost all the viruses investigated in the exotic companion mammals analyzed. Since these animals are treated as pets and always live in close contact with children, it is important to monitor and guarantee their health and the safety of their “employment” as pets.

However, even though we researched eight potential zoonotic viruses and investigated around one hundred exotic companion mammals, there are many other viruses (with unknown zoonotic potential) that, if searched for, could be found. Therefore, the level of attention must always be kept high because these animals usually make a long journey before arriving in our houses and are at high risk of contact with infectious agents.

## 5. Conclusions

In conclusion, our study underlines the importance of monitoring non-conventional pets, whose role in transmitting zoonotic agents has almost been neglected until now. Further investigation of other animals and viruses is crucial to guarantee public health, especially for children, since they are used to playing and closely interacting with these pets.

## Figures and Tables

**Figure 1 animals-14-01765-f001:**
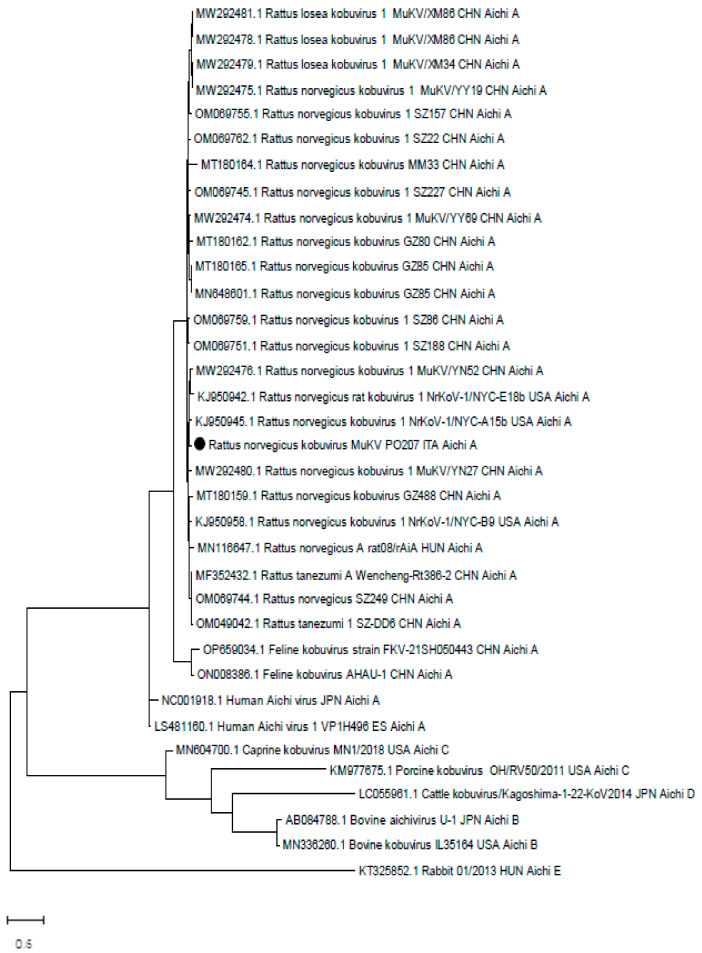
Phylogenetic tree based on the alignments of the 200 bp VP1 partial fragments. Accession numbers, host, strain name, country of detection, and species are reported for each entry. The strain identified in this study is indicated by black dots.

## Data Availability

All data supporting the present study are reported in this study. Sequence data presented in this study are openly available in the GenBank database.

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
