# Peer review of "Identification of Aichivirus in a Pet Rat (Rattus norvegicus) in Italy"

_animals, 2024, doi:10.3390/ani14121765_

Round 1

Reviewer 1 Report (Previous Reviewer 1)

Comments and Suggestions for Authors

All issues have been addressed

Author Response

Thanks for the review

Reviewer 2 Report (Previous Reviewer 2)

Comments and Suggestions for Authors

I have already reviewed the previous version of this manuscript. As I commented before, it is important to investigate the occurrence of zoonotic viruses in conventional and non-conventional pets. However the first version of the manuscript was confuse and difficult to understand. Now, the manuscript was a little bit improved. The title is more appropriate and it is possible to have a better understanding of the study after reading the Abstract. However the complete text is not ready to publication and the authors need to work hard to get a manuscript to be properly peer reviewed. I am highlighting some necessary improvements below.

First, the Introduction should present clearer the reason to investigate different viruses,  since the study aimed to search for many viruses and not only kobuviruses. I suggest to write a novel second paragraph to describe the different viruses infecting nonconventional pets. And to transfer the current second paragraph to the Discussion.

Second, the authors must select the important information for the manuscript. They do not need to report all experiments performed in the lab. In my opinion, sections “Viral recovery with murine norovirus” and “SARS-CoV-2 Droplet digital Polymerase Chain Reaction (dd-PCR)” should be removed.

Third, there are many unnecessary tables and figures for a Communication! Primers pairs and probes could be transferred to supplementary material. Data presented in Table 2 are already presented in the second paragraph of the section “Animals investigated”. It is also difficult to understand the reason to present a table with only one positive result? Tables 2 and 3 must be removed! I also suggest to remove Table 4 (or present it as supplemental material).

The Discussion is very long and presents many unnecessary comparisons. The discussion text does not need to have excessive information about different viruses and should focus on the literature review of the main experimental finding of the study (the positive result for aichivirus). Authors can use the text that was written in the Introduction and expand it a little further. They should also construct more complete paragraphs with logical sequences. And not just single sentences.

There are other structural, conceptual, and methodological problems in the manuscript. Therefore, I recommend rejecting it again, suggesting that the authors prepare it more carefully.

Comments on the Quality of English Language

 Moderate editing of English language required

Author Response

R2.1 First, the Introduction should present clearer the reason to investigate different viruses, since the study aimed to search for many viruses and not only kobuviruses. I suggest to write a novel second paragraph to describe the different viruses infecting nonconventional pets. And to transfer the current second paragraph to the Discussion.

Reply to R2.1 We thank the reviewer and agree with his suggestion. Accordingly, we reorganized the paper as requested, including further details.

R2.2 Second, the authors must select the important information for the manuscript. They do not need to report all experiments performed in the lab. In my opinion, sections “Viral recovery with murine norovirus” and “SARS-CoV-2 Droplet digital Polymerase Chain Reaction (dd-PCR)” should be removed.

Reply to R2.2 The section “SARS-CoV-2 Droplet digital Polymerase Chain Reaction (dd-PCR)” has been removed as suggested.

The section “Viral recovery with murine norovirus” can’t be removed as it was requested from the reviewer n. 1.

R2.3 Third, there are many unnecessary tables and figures for a Communication! Primers pairs and probes could be transferred to supplementary material. Data presented in Table 2 are already presented in the second paragraph of the section “Animals investigated”. It is also difficult to understand the reason to present a table with only one positive result? Tables 2 and 3 must be removed! I also suggest to remove Table 4 (or present it as supplemental material).

Reply to R2.3 Table 3 has been removed, the others are presented as supplemental material. The information presented in the table n. 2 has been requested from the reviewer n. 1 (9. Information about animals ages and sex should be included if available). The information presented in the new table n. 3 (previously table n. 4) has been provided in response to the request made by the same reviewer n. 2 during is first revision (3. Another important thing from a conceptual point of view: viruses are generally species specific and many tests used were developed for specific genotypes/lineages. This should have been better explained in the introduction and specifically defined in the Material and Methods. Just one example: There are currently several species of known circoviruses that infect a wide variety of birds and mammals and the authors used a species specific PCR. 4. The manuscript describes some very specific PCRs and others very generic. Please explain better the specificity of each assay).

R2.4 The Discussion is very long and presents many unnecessary comparisons. The discussion text does not need to have excessive information about different viruses and should focus on the literature review of the main experimental finding of the study (the positive result for aichivirus). Authors can use the text that was written in the Introduction and expand it a little further. They should also construct more complete paragraphs with logical sequences. And not just single sentences.

Reply to R2.4 We modified the text as requested.

R2.5 Comments on the Quality of English Language

Moderate editing of English language required

Reply to R2.5 The English revision will be completed onceas the manuscript will be in the final version by the MDPI English Language Editing Service.

Reviewer 3 Report (New Reviewer)

Comments and Suggestions for Authors

The manuscript entitled “Identification of Aichivirus in a pet rat (Rattus norvegicus) in Italy” demonstrates the detection of aichivirus in rodents in Italy. The study highlighted the importance of continuous monitoring pathogens among domestic animals and humans. The topic is important, but concerns are detected, please see the suggestions below.

1. the last sentence of the abstract, Further investigations are essential in order to safeguard the public health. Please be specific what investigations are needed?

2. Please re-read the draft to avoid typos, such as, line 25 the animals s, line 71 knowlwdge, line 305, inTunisia and so on. Please pay attention to these details in writing.

3. line 65, Viruses are the primary cause of acute gastroenteritis all over the world. Are there any published findings confirm this conclusion? Please cite literature.

4. the discussion is poorly written.

Comments on the Quality of English Language

English editing is needed.

Author Response

R3.1) The last sentence of the abstract, Further investigations are essential in order to safeguard the public health. Please be specific what investigations are needed?

Reply to R3.1) We agree with the reviewer for the suggestion. Accordingly, we corrected the text specifying that detection and typing of zoonotic viruses are needed to identify novel or re-emerging viruses.

R3.2) Please re-read the draft to avoid typos, such as, line 25 the animals s, line 71 knowlwdge, line 305, inTunisia and so on. Please pay attention to these details in writing.

Reply to R3.2) Done as suggested, thanks.

R3.3) Line 65, Viruses are the primary cause of acute gastroenteritis all over the world. Are there any published findings confirm this conclusion? Please cite literature.

Reply to R3.3) We agree with the referee's observation and we corrected the paper as suggested.

R3.4) The discussion is poorly written.

Reply to R3.4) We reorganized the discussion as requested.

R3.5) Comments on the Quality of English Language

English editing is needed.

Reply to R3.5) The English revision has  been  done  by the mdpi English Language Editing Service

Round 2

Reviewer 2 Report (Previous Reviewer 2)

Comments and Suggestions for Authors

The manuscript was improved again. It has a minimum level to be assessed for assessment as communication. However, the Results section deserves additional attention from the authors, as it is poorly structured and the text is not clear. The authors need to prepare a better description of the results obtained, not confusing them with the methodology. Therefore, I recommend removing the phrase “The age categories (newborn, juvenile, and adult) and sex of the exotic companion mammals investigated are listed in Table S2” (this should be moved to Materials and Methods). Additionally, authors need to review the Discussion section as it is confusing and not well ordered. I suggest first discussing all negative results (comparing with other published articles) and then describing the only positive virus with a more succinct comparison with the scientific literature (5 paragraphs is too much for a Communication).

There are still structural, conceptual and methodological problems in the manuscript. Therefore, I also suggest that authors carry out a general review and additional careful preparation of the entire manuscript.

Comments on the Quality of English Language

Moderate editing of English language required

Author Response

R2.1 First, the Introduction should present clearer the reason to investigate different viruses, since the study aimed to search for many viruses and not only kobuviruses. I suggest to write a novel second paragraph to describe the different viruses infecting nonconventional pets. And to transfer the current second paragraph to the Discussion.

Reply to R2.1 We thank the reviewer and agree with his suggestion. Accordingly, we reorganized the paper as requested, including further details.

R2.2 Second, the authors must select the important information for the manuscript. They do not need to report all experiments performed in the lab. In my opinion, sections “Viral recovery with murine norovirus” and “SARS-CoV-2 Droplet digital Polymerase Chain Reaction (dd-PCR)” should be removed.

Reply to R2.2 The section “SARS-CoV-2 Droplet digital Polymerase Chain Reaction (dd-PCR)” has been removed as suggested.

The section “Viral recovery with murine norovirus” can’t be removed as it was requested from the reviewer n. 1.

R2.3 Third, there are many unnecessary tables and figures for a Communication! Primers pairs and probes could be transferred to supplementary material. Data presented in Table 2 are already presented in the second paragraph of the section “Animals investigated”. It is also difficult to understand the reason to present a table with only one positive result? Tables 2 and 3 must be removed! I also suggest to remove Table 4 (or present it as supplemental material).

Reply to R2.3 Table 3 has been removed, the others are presented as supplemental material. The information presented in the table n. 2 has been requested from the reviewer n. 1 (9. Information about animals ages and sex should be included if available). The information presented in the new table n. 3 (previously table n. 4) has been provided in response to the request made by the same reviewer n. 2 during is first revision (3. Another important thing from a conceptual point of view: viruses are generally species specific and many tests used were developed for specific genotypes/lineages. This should have been better explained in the introduction and specifically defined in the Material and Methods. Just one example: There are currently several species of known circoviruses that infect a wide variety of birds and mammals and the authors used a species specific PCR. 4. The manuscript describes some very specific PCRs and others very generic. Please explain better the specificity of each assay).

R2.4 The Discussion is very long and presents many unnecessary comparisons. The discussion text does not need to have excessive information about different viruses and should focus on the literature review of the main experimental finding of the study (the positive result for aichivirus). Authors can use the text that was written in the Introduction and expand it a little further. They should also construct more complete paragraphs with logical sequences. And not just single sentences.

Reply to R2.4 We modified the text as requested.

R2.5 Comments on the Quality of English Language

Moderate editing of English language required

Reply to R2.5 

R2.1 First, the Introduction should present clearer the reason to investigate different viruses, since the study aimed to search for many viruses and not only kobuviruses. I suggest to write a novel second paragraph to describe the different viruses infecting nonconventional pets. And to transfer the current second paragraph to the Discussion.

Reply to R2.1 We thank the reviewer and agree with his suggestion. Accordingly, we reorganized the paper as requested, including further details.

R2.2 Second, the authors must select the important information for the manuscript. They do not need to report all experiments performed in the lab. In my opinion, sections “Viral recovery with murine norovirus” and “SARS-CoV-2 Droplet digital Polymerase Chain Reaction (dd-PCR)” should be removed.

Reply to R2.2 The section “SARS-CoV-2 Droplet digital Polymerase Chain Reaction (dd-PCR)” has been removed as suggested.

The section “Viral recovery with murine norovirus” can’t be removed as it was requested from the reviewer n. 1.

R2.3 Third, there are many unnecessary tables and figures for a Communication! Primers pairs and probes could be transferred to supplementary material. Data presented in Table 2 are already presented in the second paragraph of the section “Animals investigated”. It is also difficult to understand the reason to present a table with only one positive result? Tables 2 and 3 must be removed! I also suggest to remove Table 4 (or present it as supplemental material).

Reply to R2.3 Table 3 has been removed, the others are presented as supplemental material. The information presented in the table n. 2 has been requested from the reviewer n. 1 (9. Information about animals ages and sex should be included if available). The information presented in the new table n. 3 (previously table n. 4) has been provided in response to the request made by the same reviewer n. 2 during is first revision (3. Another important thing from a conceptual point of view: viruses are generally species specific and many tests used were developed for specific genotypes/lineages. This should have been better explained in the introduction and specifically defined in the Material and Methods. Just one example: There are currently several species of known circoviruses that infect a wide variety of birds and mammals and the authors used a species specific PCR. 4. The manuscript describes some very specific PCRs and others very generic. Please explain better the specificity of each assay).

R2.4 The Discussion is very long and presents many unnecessary comparisons. The discussion text does not need to have excessive information about different viruses and should focus on the literature review of the main experimental finding of the study (the positive result for aichivirus). Authors can use the text that was written in the Introduction and expand it a little further. They should also construct more complete paragraphs with logical sequences. And not just single sentences.

Reply to R2.4 We modified the text as requested.

R2.5 Comments on the Quality of English Language

Moderate editing of English language required

Reply to R2.5 The English revision has  been  done  by the mdpi English Language Editing Service.

This manuscript is a resubmission of an earlier submission. The following is a list of the peer review reports and author responses from that submission.

Round 1

Reviewer 1 Report

Comments and Suggestions for Authors

The manuscript "Are non conventional pets potential carriers of zoonotic viruses? A two years study" reporting the first detection of Aichivirus in a pet rat from Italy, is an interesting and novel research. However, I have some comments.

1. In the Introduction section it should be explicitly clarified if there are previous reports of virus detection in non-conventional pets from Italy. Has any of these viruses been detected before in wild or domestic animals from Italy? This should be included in the text. More background information should be incorporated to enrich the introduction.

2. Kobuvirus classification, particularly, Aichivirus species A-F should be described in the introduction.

3. I suggest to reorganize the Introduction in order to make it clearer, the paragraph stating the objective of the work could be moved to the end of this section. 

4. Were nasal and/or fecal swabs available for testing? These samples would have been an important data complement.

5. Authors should mention how were Real-Time PCRs optimized. What positive controls were used for each Real-Time PCR? In case any of the Real-Time PCR conditions were modified from the original ones reported in the references, authors should describe this point.

6. It would be important if authors could include the recovery yield of the murine norovirus added to the samples compared to the PBS control.

7. The organs in which virus detection was obtained should be included in Table 2.

8. The Ct value obtained for Aichivirus must be included in the text.

9. Information about animals ages and sex should be included if available.

10. What do authors mean with no conclusive results for the coronavirus PCR? The PCR products could not be sequenced or were they contaminated?

11. The phylogenetic tree lacks more data. Some sequences of Aichivirus species B, C, D, E and F should be included and these species must be labelled in the tree figure. 

Also, I recommend to perform the phylogenetic reconstruction with 1000 bootstraps replicates. 

What DNA substitution model was used? How was it selected? 

Was a Maximum Likelihood method also evaluated? 

12. CoV abbreviation should be defined within the text.

13. Authors could have implemented other PCRs to amplify more genomic regions of the brain sample from the pet rat positive for Aichivirus or performed an NGS study in order to obtain further data to support these findings.

Comments on the Quality of English Language

English language should be thoroughly revised throughout the manuscript.

Reviewer 2 Report

Comments and Suggestions for Authors

Manuscript " Are non conventional pets potential carriers of zoonotic viruses? A two years study” (animals-2807999)

The main aim of the study was to investigate the presence of different “potential zoonotic” viruses (according to the authors: aichivirus, sapovirus, astrovirus, hepatitis A virus, noroviruses (GI and GII), rotavirus, circovirus, coronaviruses) by Real-time PCR and end point PCR in small mammals sold as pets in pet shops of Southern Italy. The sampling included nine different species of small mammals (golden hamsters, Java squirrels, mongolian gerbils, peruvian guinea pigs,pet mice, pet rats, dwarf rabbits, african hedgehogs and sugar gliders). It was reported that 12 pet rats, 11 pet mice and 1 golden hamster tested positive to a PCR amplifying a conserved gene (RdRp) of coronavirus (however further characterizaton by sequencing gave not conclusive results??). A pet rat also resulted positive to aichivirus and its sequence showed similarity with a murine kobuvirus-1 strain identified in China. The authors state that, to their knowledge, this is the first study reporting the detection of aichivirus in rodents in Italy and suggest that this virus was probably introduced through the importation of animals from Asia.

In my opinion, this type of study is very important! Analysis of the occurrence of zoonotic viruses in conventional and non-conventional pets is necessary to monitor the circulation of any pathogenic agent (human or animal) and monitor the emergence of new microorganisms of concern to health. However, it is very difficult to understand the manuscript, as it was very poorly prepared for submission. The text is not clear and quite confusing. The authors also did not define a main objective and the title does not coincide with the rest of the article. There also some incomplete information, preventing a better understanding of the study. Finally, there are several other structural, conceptual, and methodological problems in the manuscript. Therefore, I recommend rejecting it, suggesting that the authors prepare it in a more understandable way. Here are some additional considerations to help the authors in the preparation of one scientific article which could be submitted to Animals:

1) As the authors are submitting as a communication and want to highlight the occurrence of a specific virus (achivirus), they have to define this finding as the main focus of the publication (including highlighting this discovery in the title).

2) Please avoid a hodgepodge of information about different viruses and focus on what really matters.

3) Another important thing from a conceptual point of view: viruses are generally species specific and many tests used were developed for specific genotypes/lineages. This should have been better explained in the introduction and specifically defined in the Material and Methods. Just one example: There are currently several species of known circoviruses that infect a wide variety of birds and mammals and the authors used a species specific PCR.

4) The manuscript describes some very specific PCRs and others very generic . Please explain better the specificity of each assay.

5) The authors should avoid presenting results for inconclusive findings (highlighted even in the Abstract)! Please remove them from the manuscript.

6) Please review carefully and describe in an understandable way the methods used. For example, the authors describe different methods for the SARS-CoV2 detection.

7) It is necessary to explain the sequential order of the experiments performed in the Results section. The findings are presented in tables without a better explanation in the text.

Etc.

The Discussion has also to be rewritten completely. And the manuscript should be reconsidered to publication only after the authors reorganize it completely.

Comments on the Quality of English Language

Extensive editing.